# Modified Magnesium Alkyls for Ziegler–Natta Catalysts

Julia Felicitas Schwarz [1], Thorsten Holtrichter-Rößmann [2], Claus Günter Liedtke [2], Diddo Diddens [3] and Christian Paulik [1,*]

[1] Institute for Chemical Technology of Organic Materials, Johannes Kepler University (JKU), Altenberger Strasse 69, 4040 Linz, Austria

[2] Lanxess Organometallics GmbH, Ernst-Schering-Straße 14, 59192 Bergkamen, Germany

[3] Forschungszentrum Jülich GmbH, Helmholtz-Institute Münster (IEK-12), Corrensstraße 46, 48149 Münster, Germany

**\*** Correspondence: christian.paulik@jku.at

**Abstract:** Magnesium alkyls such as butyl octyl magnesium and butyl ethyl magnesium are used as precursors for highly active and water-free magnesium chloride support materials for Ziegler–Natta catalysts. These alkyls show a high viscosity in hydrocarbon solvents which negatively affect their industrial application. Density functional theory (DFT) calculations supported the hypothesis that magnesium alkyls can form oligomeric chain structures responsible for the high viscosity. Heterocumulenes such as isocyanates, isothiocyanates and carbodiimides were studied as additives reducing the viscosity, supported by DFT calculations. The modified alkyls have further been tested in catalyst synthesis and in the polymerization of ethylene. The polymerization results showed high activities and similar polymer properties compared with a catalyst prepared without modified magnesium alkyl.

**Keywords:** magnesium alkyls; viscosity reduction; DFT calculations; heterocumulenes; Ziegler–Natta catalysts; ethylene polymerization

## 1. Introduction

Polyethylene (PE) and polypropylene (PP) are the most commonly used polymers in the world with a demand of nearly 50% of the 368 million tons of produced plastics [1].

Besides the broad application range and cheap raw materials, one of the key drivers for the success story of polyolefins are the highly optimized and efficient processes based on an extremely active and durable catalyst [2–9].

The discovery of $MgCl_2$ as a support material and activator of a classical Ziegler–Natta catalyst increased the overall activity by a factor of 10 and made the removal of the catalyst in the polyolefin product obsolete. The development of heterogenous Ziegler–Natta catalysts still remains an important and vital field of research in which the focus shifted to the modification of established catalyst systems via internal and external donors or the generation of specific magnesium chloride [4,10–14].

The synthesis of suitable $MgCl_2$ as a carrier material can be carried out in a number of different ways, but four have gained industrial significance and result in an active magnesium chloride support with a specific morphology. Magnesium chloride can be dissolved in a suitable solvent and applied for the generation of catalysts [15]. Furthermore, magnesium ethanoate in different grades and morphologies as moisture-sensitive powder can be applied to make a tailor-made and highly active support [15–18]. Additionally, highly disordered and nanostructured MgCl2 is also obtained from Mg ethoxide [18–20]. Alkyl magnesium chloride solutions supplied in ethers are also commonly used as a source for dry magnesium chloride [21,22]. The fourth elegant alternative are dialkyl magnesium compounds in aliphatic solvents combining the advantage of excluding the presence of water, ether compounds and deactivating impurities by its chemical nature and good processability as liquids compared to moisture-sensitive solids [23–26].

Most dialkyl magnesium compounds are insoluble or almost insoluble in hydrocarbon solvents due to the characteristics of the R-Mg-R units to assemble via two-center three-electron bond between the magnesium atom and the $\alpha$-carbon with another R-Mg-R unit (Figure 1) to form an organometallic polymer structure. Usually, n is very large for simple unbranched dialkyls and polymeric structures are formed [27–38]. The high viscosities at even low Mg concentrations are a macroscopic effect of the polymeric structure of the dialkyl magnesium compounds in non-donating solvents [31,34,36,37].

**Figure 1.** Polymeric structure of organomagnesium alkyls [31,36,37].

The industrially most relevant compounds are butyl ethyl magnesium (BEM), butyl octyl magnesium (BOMAG) and di-*n*-butyl magnesium, which are available in 10–20% solutions in heptane and represent the optimum between concentration (accessible magnesium), viscosity, industrial processability and economic factors. But the low concentration (low soluble Mg) and high viscosity of dialkyl magnesium solutions limit their applications in an industrial environment and make them unfavorable for further development and process investigations [36].

Nevertheless, the structural features and characteristics of the dialkyl magnesium compounds in solution, as well as their effect on viscosity have received remarkably little attention in the literature. The application of a selected modifier for the later catalyst system in combination with the dialkyl magnesium can only be found in a few examples in patent literature [27,39–41].

In our work, we have identified that heterocumulenes and especially carbodiimides reduce the viscosity of dialkyl magnesium solutions significantly and allow the synthesis of higher concentrated solutions without increasing the viscosity or reducing the catalytic performance of the subsequent Ziegler–Natta catalyst.

The reaction of magnesium dialkyl compounds with a limited number of heterocumulenes in an equimolar ratio in donating solvents such as ethers has been studied in an elegant work [42,43], but the detailed structural features of pure magnesium dialkyl compounds in hydrocarbon solution as well as the influence of the heterocumulene on the viscosity remain unexplored.

## 2. Results and Discussion

### 2.1. Viscosity of Magnesium Alkyls

The common magnesium alkyls BOMAG and BEM are commercially mainly available in heptane or toluene. Usually, an Al alkyl is used at the beginning of the Mg dialkyl synthesis to ensure a safe initiation of the reaction and to reduce the viscosity of the pure Mg alkyl. Without Al alkyl, the reaction would not start, and the Mg alkyl would be soluble in aliphatic solvents [27,36,39,40]. However, higher Al alkyl concentrations could lead to catalyst poisoning, as indicated in [44]. The viscosities of BOMAG and BEM are higher in toluene by around 45% than in heptane (Table 1).

**Table 1.** Viscosities, aluminum content and concentrations of industrially relevant magnesium alkyls in different solvents.

| Entry | Alkyl | $w_{Mg}$ (wt%) | $w_{Alkyl}$ (wt%) | $w_{Al}$ (ppm) | Solvent | $\eta$ (mPa·s) |
|-------|-------|---------------|-------------------|---------------|---------|----------------|
| 1 | BEM | 4.39 | 20.0 | 901 | Heptane [1] | 61.8 |
| 2 | BEM | 4.32 | 19.6 | 900 | toluene | 127.8 |
| 3 | BOMAG | 2.97 | 20.4 | 780 | Heptane [1] | 40.2 |
| 4 | BOMAG | 2.82 | 19.3 | 600 | toluene | 68.0 |
| 5 | di-octyl magnesium | 1.93 | 19.9 | 560 | Heptane [1] | 15.0 |
| 6 | *n*-butyl-*sec*-butyl magnesium | 5.56 | 31.7 | 17 | *n*-hexane | 2.4 |

[1] mixture of isomers.

BEM has the highest viscosity, followed by BOMAG and di-octyl magnesium. Therefore, with the increasing chain length of the alkyl group the viscosity decreases. Compared in terms of the magnesium concentration, BEM exhibits the highest magnesium content. Since these magnesium alkyls serve as a magnesium source, a high magnesium concentration is desired. The optimum between the magnesium concentration and viscosity is an important factor for the choice of magnesium alkyl.

The viscosity for *n*-butyl-*sec*-butyl magnesium with 2.4 mPa·s is, at a magnesium content of 4.32 wt%, very low; therefore, no further aluminum alkyl is added as a modifier. This is in good agreement with the density functional theory (DFT) calculations (Section 2.3), as an agglomeration of the molecules is not favored. However, the production process for this alkyl is more complex because it cannot be obtained via the reaction of alkyl halides with magnesium in hydrocarbon solvents. Thus, the synthetic route is through *sec*-butyllithium, and this is significantly more expensive [29,34,36,45,46].

*2.2. Effect of Heterocumulens on Viscosity*

Different heterocumulens from the substance groups of carbodiimides, isocyanates and isothiocyanates were used for the viscosity modification. The exact additive quantities are listed in Table S1, as well as the exact viscosity values. Isocyanates are most reactive with bases, followed by carbodiimides, isothiocyanates, carbon disulfide; $CO_2$ is the least efficient reactant. With carbon dioxide, a reaction can only take place with organometallic compounds. For isothiocyanate and isocyanate, the substituents also have an important role. Aryl-substituted heterocumulenes are usually more reactive than alkyl compounds [47].

When isocyanates were added, a reaction was detected by measuring an increase in temperature in the reaction medium of 1.7 °C for hexamethylene diisocyanate and 3.7 °C for toluene-2,4 diisocyanate. In addition, a solid precipitated immediately upon addition and therefore no viscosity reduction effect was obtained (Figure 2, Table S1). The increasing viscosity can be explained by the removal of the aluminum alkyl contained in the commercial magnesium alkyl solution with the isocyanates. Isocyanates generally tend to dimerize and trimerize and are therefore probably simply too reactive. The solid was not investigated further as this was not of interest in the context of this research [48].

With the less reactive carbodiimide compounds, no solid formation was observed, and they were able to reduce the viscosity by more than 50% at the given concentration. A temperature increase between 0 and 3.3 °C was observed when the additives were added.

The reactivities of isothiocyanates were slower and therefore a reaction time of 2 h was necessary. Within this time, a color change from colorless to yellow was observed as well as no formation of a solid. The temperature increase was between 0 and 3 °C. Only with trimethylsilyl isothiocyanate and 2,6-dimethylphenyl isothiocyanate was a viscosity drop by about 50% achieved.

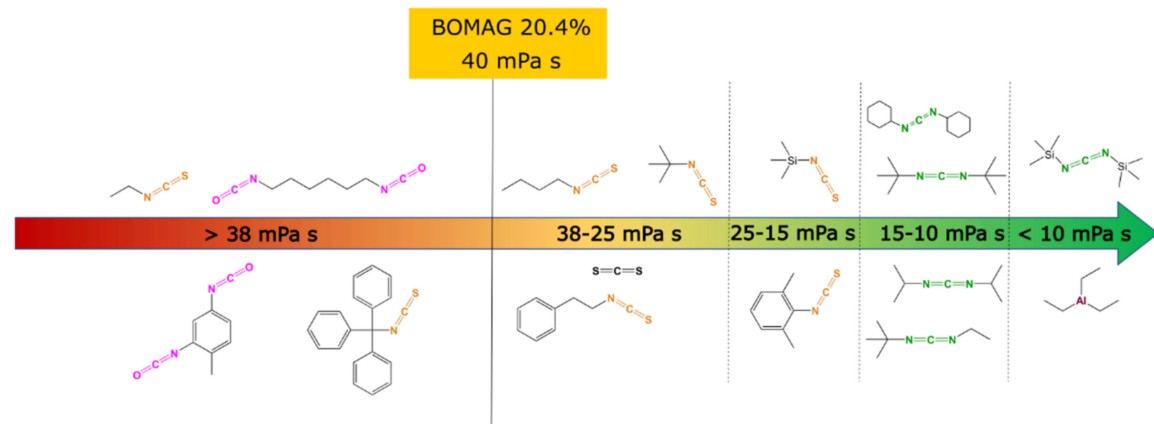

**Figure 2.** Visualization of the viscosity reduction effect due to the addition of heterocumulenes to a 20.4% butyl octyl magnesium (BOMAG) solution in heptane (mixture of isomers). The concentrations of the additives are listed in Table S1.

Triethylaluminum (TEA) was only used for comparison. Summarized, in particular the carbodiimides with alkyl substituents are highly effective viscosity reducers, trimethylsilyl carbodiimide being almost as effective as TEA.

Heptane solutions of BEM and BOMAG show an exponential increase in viscosity with increasing concentration (Figure 3a, Table S2). A commercially available BEM solution of around 20% exhibits a dynamic viscosity of 61.8 mPa·s which increases up to 600 mPa·s at a concentration of 33.2%. Similar results were observed for BOMAG but at lower viscosities. Considering the magnesium content, however, BOMAG has a higher viscosity in comparison to BEM (Figure 3b).

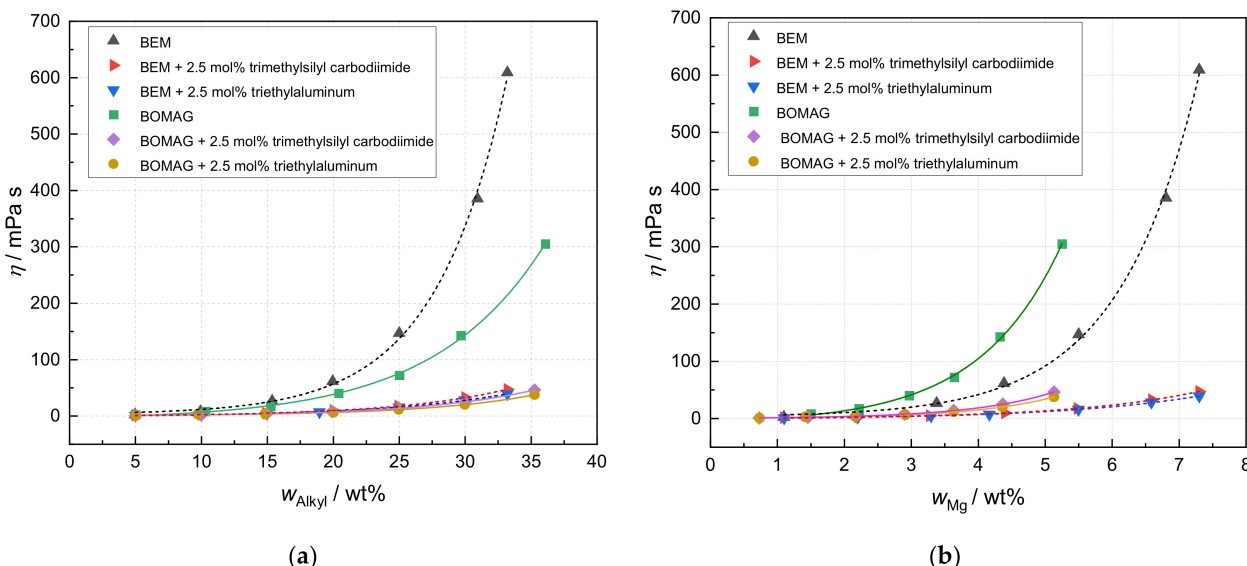

(a)  (b)

**Figure 3.** (**a**) Viscosity profiles related to the alkyl concentration for butyl ethyl magnesium (BEM) and BOMAG in heptane (mixture of isomers). Furthermore, the viscosity curves of the two alkyls with the addition of 2.5 mol% triethylaluminum and 2.5 mol% trimethylsilyl carbodiimide are displayed; (**b**) Viscosity profiles for the unmodified magnesium alkyls and modified ones related to the magnesium content.

Furthermore, the effect of the additives TEA and trimethylsilyl carbodiimide was studied regarding the viscosity at different magnesium alkyl concentrations. TEA was chosen for comparison only. The impact of reducing the viscosity by adding a 2.5 mol% fraction is almost as high for the trimethylsilyl carbodiimide as for TEA (Figure 3).

The viscosity of a 20% BEM solution decreases from 61.8 mPa·s to 9.7 mPa·s using the carbodiimide and from 609 mPa·s to 47.4 mPa·s for a 33.2% BEM solution. Thus, the concentrated 33.2% solution would still have a lower viscosity than the commercially available 20% solution. The same effect can be observed with BOMAG, where the 35.3% solution modified with trimethylsilyl carbodiimide has a viscosity of 46 mPa·s instead of 306 mPa·s and this is almost equal to the viscosity of a 20.4% solution with 40 mPa·s.

Additionally, the concentration dependence of the two additives on the solution with the highest viscosity was investigated (Figure 4, Table S3). For this purpose, the 33.2% BEM solution in heptane was used. Already with the addition of 1 mol% trimethylsilyl carbodiimide, the viscosity drops to one-quarter of the initial viscosity.

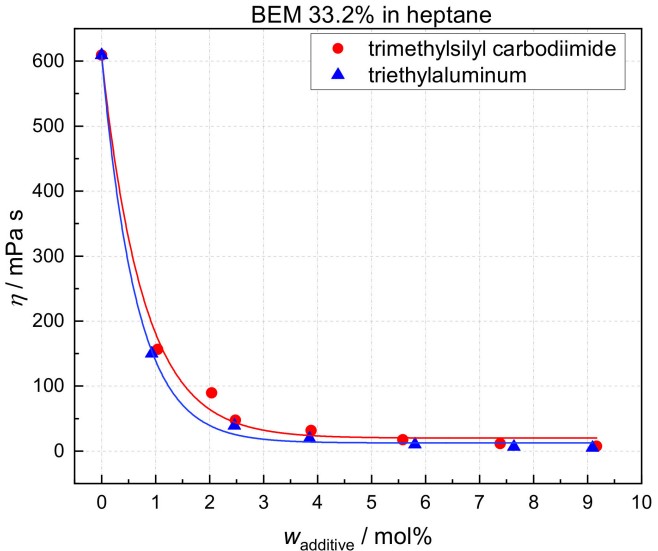

**Figure 4.** Viscosity profile of a 33.2% BEM solution in heptane with addition of different concentrations of the additives triethylaluminum (blue) and trimethylsilyl carbodiimide (red).

The viscosity reduction by the addition of approx. 2.5 mol% additive was also tested for both alkyls in toluene (Table 2). The viscosity reduction effect is comparable to the effect in heptane but at a higher total viscosity level, since the intrinsic viscosity of the magnesium alkyls in toluene is higher.

**Table 2.** Viscosities of BEM and BOMAG and the modified alkyls with triethylaluminum and silylcarbodiimide in the two solvents toluene and heptane. Additionally, the additive amount, alkyl concentration and aluminum content are listed.

| Entry | Alkyl | Additive | Solvent | $w_{Additive}$ (mol%) | $w_{Alkyl}$ (wt%) | $w_{Mg}$ (wt%) | $w_{Al}$ (ppm) | $\eta$ (mPa·s) |
|---|---|---|---|---|---|---|---|---|
| 1 | BEM | - | toluene | - | 19.64 | 4.32 | 900 | 127.8 |
| 2 | BEM | triethylaluminum | toluene | 2.67 | 19.64 | 4.32 | 900 | 14.0 |
| 3 | BEM | trimethylsilyl carbodiimide | toluene | 2.44 | 19.64 | 4.32 | 900 | 18.3 |
| 4 | BOMAG | - | toluene | - | 19.31 | 2.82 | 600 | 68.0 |
| 5 | BOMAG | triethylaluminum | toluene | 2.72 | 19.31 | 2.82 | 600 | 9.5 |
| 6 | BOMAG | trimethylsilyl carbodiimide | toluene | 2.75 | 19.31 | 2.82 | 600 | 11.8 |
| 7 | BEM | - | heptane [1] | - | 19.95 | 4.39 | 901 | 61.8 |
| 8 | BEM | triethylaluminum | heptane [1] | 2.51 | 18.95 | 4.17 | 856 | 7.23 |
| 9 | BEM | trimethylsilyl carbodiimide | heptane [1] | 2.48 | 20.01 | 4.40 | 903 | 9.68 |
| 10 | BOMAG | - | heptane [1] | - | 20.41 | 2.97 | 780 | 40.20 |
| 11 | BOMAG | triethylaluminum | heptane [1] | 2.53 | 20.02 | 2.91 | 738 | 6.33 |
| 12 | BOMAG | trimethylsilyl carbodiimide | heptane [1] | 2.52 | 19.91 | 2.91 | 734 | 7.62 |

[1] Isomeric mixture.

### 2.3. DFT Calculations

Density functional theory (DFT) calculations were carried out to obtain energetic impacts of the magnesium alkyls and the magnesium alkyl additive interactions.

To keep the calculations computationally expedient, we focused on dimethyl, diethyl, dibutyl and *sec*-butyl magnesium, ignoring the experimentally employed octyl species. Here, the variation of the alkyl length allows us to study its dependence on the resulting formation energies for magnesium alkyl oligomers or complexes with an additive molecule. For additives, we focused on dimethyl and diethyl carbodiimide as references as well as the experimentally relevant compounds trimethylsilyl carbodiimide and *N,N′*-dicyclohexyl carbodiimide. TEA was used as a reference additive.

In a first step, we studied the energetic stability of linear oligomers as depicted in Figure 5.

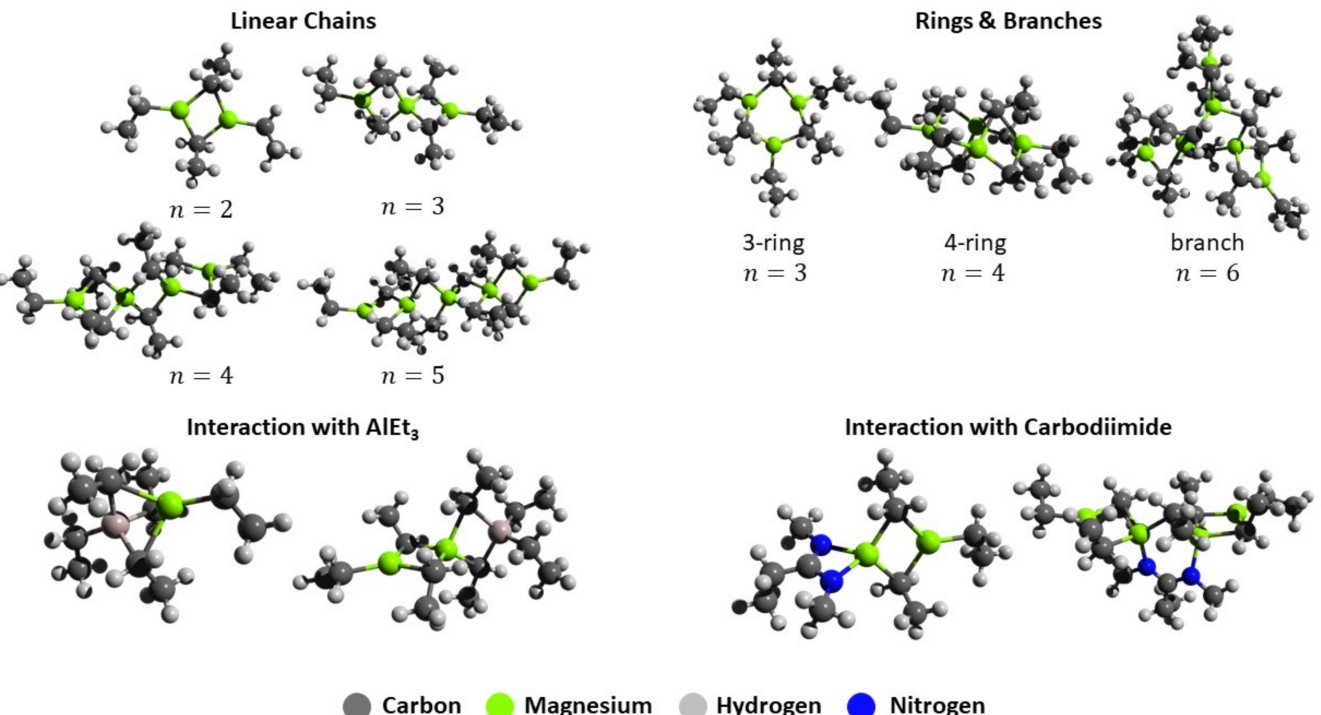

**Figure 5.** Visualization of the different magnesium alkyl oligomers and their clusters with additive molecules that were studied in this work.

The corresponding oligomerization energies $\Delta E$ and free energies $\Delta G$ per monomer are listed in Table 3 (i.e., $\Delta E/n$ and $\Delta G/n$, with n being the number of monomers). We note that while even the dimerization shows negative values for $\Delta E/n$ and $\Delta G/n$, these energies become smaller until they saturate at about $n \approx 4$–5. Similar values are found for all substituents (methyl, ethyl, butyl and *sec*-butyl). In total, these findings show that (a) the formation of longer magnesium alkyl polymers is thermodynamically favored and thus rationalizes the experimentally observed viscosity increase and that (b) our chosen model system is sufficiently large to qualitatively reproduce the most salient features.

Of course, linear chains are not the only possible form of aggregates. For example, magnesium alkyls may also form ring-like oligomers or branches within linear chains [49]. Therefore, we performed additional calculations of rings composed of three and four monomers, respectively, as well as a branched chain (see Figure 5).

From the resulting formation (free) energies in Table 3 we observe that all investigated structures also show negative values for $\Delta E/n$ and $\Delta G/n$, which are comparable in magnitude to the respective values for linear chains.

**Table 3.** Formation energies and free energies per monomer, $\Delta E/n$ and $\Delta G/n$, for the possible aggregates depicted in Figure 5. The energies are shown for the formation of linear chains and both the other aggregation ways, the ring-like oligomer formation or branches within linear chains.

| | | Methyl | | Ethyl | | Butyl | | *n*-Butyl *sec*-Butyl | |
|---|---|---|---|---|---|---|---|---|---|
| Entry | *n* | $\Delta E/n$ | $\Delta G/n$ | $\Delta E/n$ | $\Delta G/n$ | $\Delta E/n$ | $\Delta G/n$ | $\Delta E/n$ | $\Delta G/n$ |
| | | | | **Linear Chains** | | | | | |
| 1 | 2 | −8.7 | −2.5 | −9.7 | −3.4 | −10.6 | −2.4 | −7.7 | −0.6 |
| 2 | 3 | −13.2 | −5.5 | −14.7 | −5.2 | −15.8 | −5.3 | - | - |
| 3 | 4 | −15.2 | −5.9 | −17.1 | −6.3 | −18.5 | −6.5 | - | - |
| 4 | 5 | −16.2 | −5.9 | −17.6 | −5.4 | −19.8 | −7.0 | −15.5 | −2.6 |
| | | | | **Rings and Branches** | | | | | |
| 5 | 3 | −7.6 | 1.0 | −9.9 | 0.1 | −10.8 | 0.3 | −11.0 | −0.3 |
| 6 | 4 | −15.2 | −5.9 | −16.3 | −5.4 | −17.9 | −6.0 | −16.6 | −4.5 |
| 7 | 5 | −15.4 | −4.9 | −17.5 | −5.2 | −19.9 | −6.1 | −15.4 | −1.2 |

Hence, we conclude that both the formation of long linear chains as well as their branching or the formation of smaller rings seems likely from a thermodynamic standpoint, suggesting a mixture of all these species under experimental conditions.

However, we observe from additional calculations with *n*-butyl *sec*-butyl magnesium oligomers that $\Delta E/n$ and $\Delta G/n$ become noticeably larger, presumably due to the larger steric strain. Thus, we expect the above-mentioned tendencies to be somewhat weaker for *n*-butyl *sec*-butyl magnesium as compared to BOMAG. This is also consistent with the experimental data, as *n*-butyl *sec*-butyl has much lower viscosities than BOMAG.

Finally, we focus on the question how the presence of additive molecules affects the stability of magnesium alkyl polymers. In particular, we focus on two different ways in which a carbodiimide can interact with the alkyl magnesium chain: first, we studied its termination by forming a 4-numbered ring structure and second, its intercalation to the chain to form a 6-ring structure (Figure 5). The corresponding energies are shown in Table 4. In all cases, there was a significantly negative interaction, demonstrating the strong interaction between carbodiimide additive and magnesium alkyl. Apparently, the intercalation mechanism is more stabilized compared to the termination mechanism, of which only the latter would lead to significantly shorter polymer chains in the experiments. However, one might speculate that the intercalation can subsequently lead to the cleavage of the chain as an intermediate to chain termination, which would explain the lowered viscosity observed experimentally.

**Table 4.** Interaction energies and free energies $\Delta E$ and $\Delta G$ for the alkyl magnesium-additive complexes depicted in Figure 5. The values for the formation of 4-numbered ring structures and the intercalation to the chain to form 6-ring structures are shown.

| | | Methyl | | | | Ethyl | | | | Butyl | | | |
|---|---|---|---|---|---|---|---|---|---|---|---|---|---|
| | | 4-Ring | | 6-Ring | | 4-Ring | | 6-Ring | | 4-Ring | | 6-Ring | |
| Entry | Carbodiimide | $\Delta E$ | $\Delta G$ | $\Delta E$ | $\Delta G$ | $\Delta E$ | $\Delta G$ | $\Delta E$ | $\Delta G$ | $\Delta E$ | $\Delta G$ | $\Delta E$ | $\Delta G$ |
| 1 | Me | −76.5 | −50.9 | −118.7 | −68.3 | −85.0 | −54.7 | −138.8 | −76.2 | −83.8 | −56.1 | −132.5 | −73.7 |
| 2 | Et | −75.8 | −51.1 | −116.9 | −64.2 | −85.7 | −127.3 | - | - | −83.6 | −53.9 | −132.7 | −73.9 |
| 3 | Si(Me)$_3$ | −61.0 | −35.1 | −101.2 | −48.1 | −71.9 | −39.7 | −123.2 | −59.0 | −70.1 | −40.3 | −116.3 | −56.7 |
| 4 | Cyclohexyl | −72.0 | −45.1 | −119.7 | −67.1 | −81.1 | −48.6 | −140.2 | −75.1 | −79.4 | −48.8 | - | - |

Overall, the variation with the substituent at the carbodiimide is rather small. However, the interaction with the carbodiimide is still substantially larger than the corresponding interaction with the conventional additive TEA (Table 5).

**Table 5.** Interaction energies and free energies $\Delta E$ and $\Delta G$ for the alkyl magnesium-aluminum complexes depicted in Figure 5.

| Entry | | $\Delta E$ | $\Delta G$ | $\Delta E$ | $\Delta G$ |
|---|---|---|---|---|---|
| | **R** | $(MgR_2)$ $(AlEt_3)$ | | $(MgR_2)_2$ $(AlMe_3)$ | |
| 1 | Methyl | −21.5 | −8.9 | −16.7 | 9.5 |
| 2 | Ethyl | −25.1 | −10.4 | −48.4 | −18.9 |
| 3 | Butyl | −27.3 | −11.4 | −53.0 | −20.9 |

*2.4. Ziegler–Natta Catalyst Synthesis and Polymerization Experiments*

To ensure that the additives are no catalyst poison in the further process, Ziegler–Natta catalysts were prepared, which were then tested in ethylene polymerization. The catalyst synthesis was based on the precipitation process, where magnesium chloride is precipitated first, followed by titanation [24,50,51].

For the catalyst synthesis and for the polyerization experiments, the additives with the best viscosity effect were selected from the carbodiimide and isothiocyanate groups. For both classes, the trimethylsilyl compounds achieved the best effect. For comparison, the two compounds without silicon were used, i.e., *tert*-butyl carbodiimide and *tert*-butyl isothiocyanate.

The titanium content for the catalysts with silicon is considerably higher (by about 3 wt%) than the reference catalyst and catalysts using additives without silicon (Table 6).

**Table 6.** Titanium content and polymerization activities related to the catalyst amount and the titanium content for the synthesized catalysts using the modified magnesium alkyls.

| Entry | Additive | $w_{Additive}$ (wt%) | $w_{Ti}$ (wt%) | $R_p$ ($kg_{PE}·g_{Cat}^{-1}·h^{-1}$) | $R_{p,Ti}$ ($kg_{PE}·g_{Ti}^{-1}·h^{-1}$) |
|---|---|---|---|---|---|
| 1 | no additive | 0 | 7.4 | 25.4 ± 0.982 | 343.5 ± 13.26 |
| 2 | trimethylsilyl carbodiimide | 2.57 | 10.83 | 33.1 ± 0.006 | 305.1 ± 0.057 |
| 3 | *tert*-butyl carbodiimide | 2.63 | 7.62 | 26.5 ± 0.460 | 347.3 ± 6.034 |
| 4 | trimethylsilyl isothiocyanate | 2.48 | 11.06 | 24.6 ± 3.317 | 222.9 ± 29.99 |
| 5 | *tert*-butyl isothiocyanate | 2.78 | 7.85 | 24.3 ± 1.639 | 309.8 ± 20.89 |

The catalysts with similar titanium content provide very good and similar polymerization results with an activity related to the titanium ($R_{p,Ti}$) of 347 $kg_{PE}·g_{Ti}^{-1}·h^{-1}$ for *tert*-butyl carbodiimide and 310 $kg_{PE}·g_{Ti}^{-1}·h^{-1}$ for *tert*-butyl isothiocyanate compared to the standard catalyst with 344 $kg_{PE}\ g_{Ti}^{-1}\ h^{-1}$.

Trimethylsilyl isothiocyanate has an equal activity ($R_p$) based on the amount of catalyst, but due to the high titanium content, the activity based on this is significantly lower. The catalyst amount activity for trimethylsilyl carbodiimide is quite high with 33 $kg_{PE}·g_{Cat}^{-1}·h^{-1}$ compared to the standard catalyst with 25.4 $kg_{PE}·g_{Cat}^{-1}·h^{-1}$. However, because of the high titanium content, the activity related to it is lower than that of the standard system. Generally, titanium can have three oxidation states in Ziegler−Natta catalyst systems, $Ti^{2+}$, $Ti^{3+}$ and $Ti^{4+}$. The $Ti^{4+}$ is reduced by the aluminum alkyl, because this state is not active in the polymerization process, due to a strong complexation effect by Lewis bases. For the ethylene polymerization, both oxidation states $Ti^{2+}$ and $Ti^{3+}$ are suitable; for propylene, only $Ti^{3+}$. Recent studies showed that there is a copresence of active isolated $Ti^{3+}$ and $TiCl_3$ like clusters which are not active in the polymerization process [52]. In addition, $Ti^{2+}$ is inferior in relation to the catalyst activity [52–56]. Such high titanium amounts with low activity are also described in the literature because of the formation of inactive $TiCl_3(OEt)$ or $Cl_3Ti$-O-$TiCl_3$ [57,58]. Therefore, it is possible that the titanium content is in this case too high to make the best use of the active centers and it may therefore be useful to adjust the titanation amounts to achieve lower titanium values for high activity results. However, catalyst production by

the precipitation method works and the polymerization results support the conclusion that there is no significant catalyst poisoning by the tested additives.

Comparing the molecular weight distributions of the produced polyethylens obtained by high temperature-size exclusion chromatography (HT-SEC) shows that there are no major deviations between the results obtained for the standard catalyst in terms of the number average $M_n$, average molar mass $M_w$, and the dispersity index Đ (Table 7, Figure S1) compared with catalysts with a modified support. Only for *tert*-butyl isothiocyanate is a small shift to higher molar masses and narrower Đ observed.

**Table 7.** Molecular weight distribution for the obtained polymers measured with HT-SEC.

| Entry | Additive | $M_n$ (g·mol$^{-1}$) | $M_w$ (g·mol$^{-1}$) | Đ |
|---|---|---|---|---|
| 1 | no additive | 35,900 ± 4680 | 234,333 ± 27,107 | 6.5 ± 0.13 |
| 2 | trimethylsilyl carbodiimide | 37,200 ± 282 | 227,150 ± 899 | 6.1 ± 0.12 |
| 3 | *tert*-butyl carbodiimide | 35,500 ± 5939 | 214,150 ± 316 | 6.0 ± 0.13 |
| 4 | trimethylsilyl isothiocyanate | 39,533 ± 7182 | 241,433 ± 44,831 | 6.1 ± 0.03 |
| 5 | *tert*-butyl isothiocyanate | 49,600 ± 0 | 281,950 ± 24,745 | 5.7 ± 0.05 |

## 3. Materials and Methods

### 3.1. Magnesium Alkyl Modification

All reactions were performed under inert conditions using the Schlenk technique in argon (5.0, Linde) atmosphere or in a nitrogen-filled M. Braun glove box (<1 ppm $O_2$ and <1 ppm $H_2O$). The magnesium alkyls butyl octyl magnesium and butyl ethyl magnesium (BEM) in heptane (isomer mixture) or toluene were provided by LANXESS Organometallics GmbH and *n*-butyl *sec*-butyl magnesium (0.7 mol·L$^{-1}$ in *n*-hexane) was obtained from Sigma Aldrich. The additives (listed in Table S1) were purchased from commercial sources and used without further purification; trimethylsilyl carbodiimide was supplied by AlzChem Group AG.

For the modification around 20 g (24.5 mmol, exact values Table S1) of 20.4% BOMAG solution in heptane were used and approximately 2.5 mol% (related to the Mg content) additive were added at room temperature. After stirring for 10 min, the temperature was raised to 50 °C for half an hour (or 2 h for isothiocyanates).

After cooling down to room temperature, the viscosity was measured using a DV2T Brookfield spindle viscosimeter. The measurement was performed under an argon stream at 21 °C and 50 rpm with a SC4-18 spindle.

Higher concentrated magnesium alkyl solutions were obtained by distilling off the heptane and the magnesium, and aluminum concentrations were determined by ICP-MS (Thermo Scientific X Series 2, Waltham, MA, USA).

### 3.2. DFT-Calculations

All DFT calculations were performed using ORCA 4.1.2 software [59]. The initial molecular structures were generated using Avogadro [60]. Subsequently, these structures were optimized via the TPSS functional [61] with a def2-TZVP basis [62] and Grimme's D3 empirical dispersion interaction [63]. To describe the impact of the surrounding solvent, the implicit CPCM solvation model [64] was used with built-in parameters for hexane. Vibrational frequencies were also computed at this level of theory to compute thermochemical contributions based on the unscaled frequencies. Following the geometry optimization, single-point calculations were carried out at the PWPB95-D3/def2-TZVP level of theory [65].

### 3.3. Zieger–Natta Catalyst Preparation

25.56 mmol 2-ethyl hexanol (98%, Merck) were added over 40 min to 12.78 mmol BOMAG 35.6% in *n*-heptane while maintaining the temperature between 0–5 °C. This alcoholate solution was subsequently added to an ethyl aluminum dichloride solution (12.78 mmol, 25.3% in heptane (mixture of isomers), LANXESS) at 60 °C, resulting in the

precipitation of magnesium chloride. After 1 h stabilization time, the resulting magnesium chloride was separated in a centrifuge and washed twice with 2 mL *n*-heptane. The $MgCl_2$ carrier was diluted with 3 mL *n*-heptane and heated to the titanation temperature of 80 °C. With the addition of $TiCl_4$ (6.30 mmol, ca. 19% in toluene, TCI) over 30 min and 40 min stabilization time, a solid brownish/red colored catalyst was obtained. The color is an indication that there is already reduced titanium in the pre-catalyst, likely in form of $TiCl_3$ clusters [53]. After washing three times with 1 mL *n*-heptane, the catalyst was dried at room temperature.

The titanium content was measured with ICP-MS. For that purpose, 3.5 mL $HNO_3$ (69%, Honeywell), 1.5 mL HCl (analpur, Analytica) and 3 mL HF (48%, Analytica) were added to approx. 0.1 g catalyst sample and decomposed using a MARS 6 digestion unit at 180 °C for 20 min. The dilutions were made with 18 MΩ water containing 1% $HNO_3$ and 0.5% HF.

*3.4. Ethylen Polymerization Experiments*

The setup, purification and procedure were based on the literature with some minor modifications [66,67]. A schematic illustration of the reactor setup is shown in Figure S2.

The following gases and liquids were used for the polymerization experiments: propane (3.5, Gerling), hydrogen (5.0, Linde), ethylene (3.0, Linde), nitrogen (5.0, Linde) and *n*-heptane (97%, Roth). With a purification unit containing oxidizing/reducing catalyst systems and molecular sieves it was ensured that no impurities were present in the reactor system. The polymerization was done in the propane as a solvent and the triethyl aluminum (TEA, Lanxess) as a cocatalyst, whereby the Al:Ti ratio was 100:1. The catalyst and cocatalyst were separately suspended in oil (Shell Ondina Öl 933) and injected manually with the help of a Gilson pipette. After a pre-contacting time of 5 min the catalyst and cocatalyst were flushed into the reactor system with *n*-heptane, which already contained 50 mg $H_2$ and 8 g ethylene. Hydrogen was used to control the molecular weight of the generated polyethylene. During the heat-up rate to 70 °C a pre-polymerization was performed. After reaching this temperature, the polymerization was continued for 30 min at constant pressure of 32 bar guaranteed by feeding ethylene through a mass flow controller into the reactor.

The obtained polymer was dried at 70 °C for one day and afterwards analyzed by high temperature-size exclusion chromatography (HT-SEC, PolymerChar, column PLgel Olexis mixed-bed 7 × 300 mm from Agilent, detector IR5) to determine the molecular weight distribution, the average molar mass $M_w$, the number average $M_n$ and the dispersity index Đ. To 2 mg of the samples 20 μL tracer (1:10 mixture of *n*-heptane: 1,2,4-trichlorobenzene) were added. The sample was afterwards dissolved in 8 mL 1,2,4-trichlorobenzene at 160 °C for 120 min and afterwards measured.

## 4. Conclusions

We herein had a detailed view of the effect of the viscosity of modified magnesium alkyl solutions on the preparation and polymerization performance of Ziegler–Natta catalysts. The DFT calculations showed that the formation of longer magnesium alkyl oligomers is thermodynamically preferable and leads to higher viscosities. It also showed that the use of additives can reduce the likelihood of oligomer formation and thus is a means to reduce the viscosity of magnesium alkyl solutions.

Heterocumulenes and especially carbodiimides can significantly reduce the viscosity. The addition of 2.5 mol% of trimethylsilyl carbodiimide results in a viscosity reduction of more than 50% of a 20.4% BOMAG solution. As a result, BOMAG could be concentrated up to approximately 35% showing the same viscosity as a commercial BOMAG solution in heptane (20.4%, 40 mPa·s). With the modified magnesium alkyl, it was still possible to synthesize a Ziegler–Natta catalyst using a state-of-the-art precipitation method. In addition, the polymerization results showed high activities and similar polymer properties compared with a catalyst prepared without modified magnesium alkyl. This could be a

new way of affording higher concentrated magnesium alkyl solutions without affecting the synthesis and performance of standard Ziegler–Natta catalysts.

**5. Patents**

Parts of this work are in patent application (WO2021233930A1) [68].

**Supplementary Materials:** The following supporting information can be downloaded at: https://www.mdpi.com/article/10.3390/catal12090973/s1, Table S1: Values and observations for the modification of the magnesium alkyl butyl octyl magnesium (20.4% in heptane) with heterocumulenes; Table S2: Viscosity values for butyl ethyl magnesium (BEM) and butyl octyl magnesium (BOMAG) modified and not modified at different alkyl concentrations. The graphic can be seen in Figure 3 of the article; Table S3: Data for Figure 4 for the viscosity profile of a 33.2% BEM solution in heptane and different additive concentrations; Figure S1: Molecular weight distribution for the obtained polymers measured with HT-SEC; Figure S2: Schematic drawing of the 0.5 L reactor setup used for the polymerization experiments.

**Author Contributions:** Conceptualization: J.F.S., C.P. and T.H.-R.; methodology: J.F.S., T.H.-R., C.P. and D.D.; validation: J.F.S., C.G.L. and D.D.; formal analysis: J.F.S., C.G.L. and D.D.; investigation: J.F.S., C.G.L. and D.D.; resources: C.P., T.H.-R. and D.D.; writing—original draft preparation, J.F.S., T.H.-R. and D.D.; writing—review and editing: C.P.; visualization: J.F.S., T.H.-R. and D.D.; supervision: C.P. and T.H.-R.; project administration: C.P. and T.H.-R.; funding acquisition: T.H.-R. and C.P. All authors have read and agreed to the published version of the manuscript.

**Funding:** This research was funded by the Austrian Research Promotion Agency (FFG), grant number 877172, Functionalized and functional metal alkyls as modifying precursors for heterogeneous polyolefin catalysis (FunAlkyl).

**Data Availability Statement:** Not applicable.

**Acknowledgments:** We would like to thank Thomas Günther of the company AlzChem Group AG for providing the trimethylsilyl carbodiimide. Furthermore, we would like to thank the technicians Karl Schütz, DI Daniel Pernusch, DI Lukas Göpperl and Gerold Rittenschober for help with any problems with the reactor system.

**Conflicts of Interest:** The authors declare no conflict of interest.

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
