# Peer review of "Modified Magnesium Alkyls for Ziegler–Natta Catalysts"

_catalysts, doi:10.3390/catal12090973_

Round 1

Reviewer 1 Report

See the attached.

Author Response

Review 1:

Comment on the manuscript entitled “Modified Magnesium Alkyls for
Ziegler-Natta Catalysts” for publication in Catalysts.
The paper focuses on the synthesis of Ziegler-Natta catalysts using
different magnesium alkyls as precursor. The paper does merit publication
after clarifying some points as follows;

  1. There were only 5 catalysts (Table 6) used in ethylene polymerization.
    Basically, they were prepared with different additives. Other characteristics
    of these catalysts should be determined such as morphology, surface area,
    and distribution of Ti. and What is the relationship between the size of catalyst and the size of
    polymer? The particle size distribution for catalysts and polymer should be

Answer: Many thanks for your comments. The main purpose of the polymerization experiments was to demonstrate that the heterocumulenes do not act as a catalyst poison. The aim is to show that it has no or almost no effect on a basic polymerization experiment. The characteristics of the polymer in terms of PDI as well as the catalyst morphology and differences are part of a new manuscript already in preparation. We think it also would mislead the reader from the core of the present paper. We already did perform some XRD experiments with the standard catalyst without additive and a catalyst with dicyclohexyl carbodiimide and they showed no difference between the two catalysts and also hat they are amorphous.

  1. As shown in Table 6, why did some additives result in the higher Ti
    content? Please clarify

Answer: Many thanks for your comments. This is so far unclear for the authors and surprising as well and part of ongoing investigations. This effect is an interesting point for further industrial improvement of catalysts.

  1. English should be polished. For instance,

Line 14 effects to affects
Line 58-59 “The industrially most relevant compounds are butyl ethyl
magnesium (BEM), butyl-58 octyl magnesium (BOMAG)”. The use of
nomenclature is not uniform (i.e. BEM and BOM).
etc.

Answer: BOMAG is the registered trade name for butyl octyl magnesium and the industrial standard for this particular compound. Lanxess is one of the main manufacturer. The product is also registered under the trade name BOMAG. To meet industrial standard and avoid confusion and misleading interpretation by the readers of the Catalyst journal we suggest to keep the wording BOMAG.

Reviewer 2 Report

The manuscript by Schwarz et al. is a complete and well-organized investigation of a class of Ziegler-Natta catalysts prepared from Mg alkyl solutions for ethylene polymerization. It is of sure interest for the readers of Catalysts, both from academia and from companies, although some issues should be addressed before it can be suitable for publication.

First of all, the authors should clarify better the advantages of heterocumulene additives onto the catalytic polymerization process. Indeed, they claim that the additives reduce the viscosity allowing a higher BOMAG concentration: so which is the actual advantage? Is it just a matter of how much ZN catalyst can be produced in the same batch? Just looking at Tables 6 and 7 there is not a net improvement in using the additives, neither considering the polymerization activities nor considering the properties of the produced polymers.

Caption of all the Tables should be expanded, describing more in detail the reported values (and caption of Figure 2 has some informatic problems).

Could the authors provide some information about structure and morphology of MgCl2 produced with their procedure? It would be useful to know at least the dimension of the particles or the specific surface area…

When the Ziegler-Natta catalyst is produced, it is a brownish/red colored solid. This is a clear evidence of the presence of some reduced Ti already in the pre-catalyst, likely in the form of TiCl3-like clusters (10.1021/acscatal.1c01735).

In Section 2.4 the authors present some possible species for the activated ZN catalysts. According to recent studies on the electronic properties of the Ti sites, on TEA-activated catalysts there is usually the copresence of active isolated Ti3+ sites and of TiCl3-like clusters not active in olefin polymerization (10.1021/acscatal.1c01735), whose presence in this case starts likely in the pre-catalysts.

In the introduction, it is worth adding that in industrial practice highly disordered and nanostructured MgCl2 is obtained not only from Mg ethanoate but also from Mg ethoxide (as published for instance in 10.1021/acscatal.1c03067, 10.1016/j.jcat.2020.06.030 and 10.1021/acs.iecr.8b05296).

Pictures in Figure 5 should be enlarged, and a legend of atoms should be added.

Why is TEA studied as additive for BEM and BOMAG? In the catalytic process TEA is added after the conversion of Mg alkyls into MgCl2 and the formation of the pre-catalyst with TiCl4, isn’t it?

Author Response

Review 2:

The manuscript by Schwarz et al. is a complete and well-organized investigation of a class of Ziegler-Natta catalysts prepared from Mg alkyl solutions for ethylene polymerization. It is of sure interest for the readers of Catalysts, both from academia and from companies, although some issues should be addressed before it can be suitable for publication.

  1. First of all, the authors should clarify better the advantages of heterocumulene additives onto the catalytic polymerization process. Indeed, they claim that the additives reduce the viscosity allowing a higher BOMAG concentration: so, which is the actual advantage? Is it just a matter of how much ZN catalyst can be produced in the same batch? Just looking at Tables 6 and 7 there is not a net improvement in using the additives, neither considering the polymerization activities nor considering the properties of the produced polymers.

We thank the reviewer for the comment. In our opinion the advantage is threefold:

  1. the customer can use existing equipment in the production plant but increase the capacity by 50 % without additional investment cost. The properties of the manufactured catalyst AND polymer remains the same. No change is always a big benefit for commercial plants
  2. For the producer the production AND shipment of a 30 % solution compared to a 20 % solution reduces logistic complexity (lesser handling of specialized containers, lesser cleaning of these containers) and reduces logistic costs.
  3. It is also a starting point to show evidence that catalyst modifications can be done by manipulating the precursor material or not. The door for more development and modification is open.

  1. Caption of all the Tables should be expanded, describing more in detail the reported values (and caption of Figure 2 has some informatic problems).

Many thanks, the tables have been described in more detail and the informatic problem is corrected.

  1. Could the authors provide some information about structure and morphology of MgCl2 produced with their procedure? It would be useful to know at least the dimension of the particles or the specific surface area…

Many thanks for the comment, the main purpose of the polymerization experiments is to demonstrate that the heterocumulenes do not act as a catalyst poison. The aim is to show that it has no or almost no effect on a basic polymerization experiment. The characteristics of the polymer in terms of PDI as well as the catalyst morphology and differences are part of a manuscript already in preparation. It would also mislead the reader from the core of the paper. We already did XRD experiments with the standard catalyst without additive and a catalyst with dicyclohexyl carbodiimide and they showed no difference between the two and that they are amorphous.

  1. When the Ziegler-Natta catalyst is produced, it is a brownish/red colored solid. This is a clear evidence of the presence of some reduced Ti already in the pre-catalyst, likely in the form of TiCl3-like clusters (10.1021/acscatal.1c01735).

Thank you for this comment, this has been added in paragraph 3.3

  1. In Section 2.4 the authors present some possible species for the activated ZN catalysts. According to recent studies on the electronic properties of the Ti sites, on TEA-activated catalysts there is usually the copresence of active isolated Ti3+ sites and of TiCl3-like clusters not active in olefin polymerization (10.1021/acscatal.1c01735), whose presence in this case starts likely in the pre-catalysts.

Thank you for this comment, this has been adapted in section 2.4.

  1. In the introduction, it is worth adding that in industrial practice highly disordered and nanostructured MgCl2 is obtained not only from Mg ethanoate but also from Mg ethoxide (as published for instance in 10.1021/acscatal.1c03067, 10.1016/j.jcat.2020.06.030 and 10.1021/acs.iecr.8b05296).

We thank the reviewer for the valuable input and expanded the introduction with the Mg ethoxide and the sources.

  1. Pictures in Figure 5 should be enlarged, and a legend of atoms should be added.

Figure 5 is enlarged and the legend of atoms are added.

  1. Why is TEA studied as additive for BEM and BOMAG? In the catalytic process TEA is added after the conversion of Mg alkyls into MgCl2 and the formation of the pre-catalyst with TiCl4, isn’t it?

Thank you for your comment, we amended the text accordingly. Usually, an Al alkyl is used at the beginning of the Mg dialkyl synthesis to ensure a safe initiation of the reaction and to reduce the viscosity of the pure Mg alkyl. Without Al alkyl the reaction would not start nor the Mg alkyl would be soluble in aliphatic solvents. (Patent US3,737,393Patent, US 4,547,477). In our catalyst preparation Al alkyls are used but several different applications use other Cl-sources and react very sensitive towards a higher Al content (Patent US5145600A). So, a neutral additive is of great importance.

Reviewer 3 Report

The authors found that heterocumulenes especially the carbodiimides could reduce the viscosity of magnesium alkyls by more than 50%,which had the same ability of aluminum alkyls. Although the heterocumulenes have no negative effects on the prepared catalysts, but they show little superiority over aluminum alkyls in effect and price.

Author Response

Review 3:

  1. The authors found that heterocumulenes especially the carbodiimides could reduce the viscosity of magnesium alkyls by more than 50%,which had the same ability of aluminum alkyls. Although the heterocumulenes have no negative effects on the prepared catalysts, but they show little superiority over aluminum alkyls in effect and price.

Thank you for your comment, we amended the text accordingly. Usually, an Al alkyl is used at the beginning of the Mg dialkyl synthesis to ensure a safe initiation of the reaction and reduce the viscosity of the pure Mg alkyl. Without Al alkyl the reaction would not start nor the Mg alkyl would be soluble in aliphatic solvents. (Patent US3,737,393Patent, US 4,547,477). In our catalyst preparation Al alkyls are used but several different applications use other Cl sources and react very sensitive towards a higher Al content (Patent US5145600A). So, a neutral additive is of great importance.

Round 2

Reviewer 3 Report

the revised manuscript can be accepted.